# Sinogram Inpainting with Generative Adversarial Networks and Shape Priors

Emilien Valat [1,*] , Katayoun Farrahi [2] and Thomas Blumensath [3]

1 Cambridge Image Analysis Group, Department of Applied Mathematics and Theoretical Physics, Centre for Mathematical Sciences, University of Cambridge, Wilberforce Rd., Cambridge CB3 0WA, UK
2 Vision, Learning and Control Group, Department of Electronics and Computer Science, University of Southampton, University Rd., Southampton SO17 1BJ, UK
3 Institute of Sound and Vibration Research, Department of Engineering and the Environment, University of Southampton, University Rd., Southampton SO17 1BJ, UK
* Correspondence: ev373@cam.ac.uk

**Abstract:** X-ray computed tomography is a widely used, non-destructive imaging technique that computes cross-sectional images of an object from a set of X-ray absorption profiles (the so-called sinogram). The computation of the image from the sinogram is an ill-posed inverse problem, which becomes underdetermined when we are only able to collect insufficiently many X-ray measurements. We are here interested in solving X-ray tomography image reconstruction problems where we are unable to scan the object from all directions, but where we have prior information about the object's shape. We thus propose a method that reduces image artefacts due to limited tomographic measurements by inferring missing measurements using shape priors. Our method uses a Generative Adversarial Network that combines limited acquisition data and shape information. While most existing methods focus on evenly spaced missing scanning angles, we propose an approach that infers a substantial number of consecutive missing acquisitions. We show that our method consistently improves image quality compared to images reconstructed using the previous state-of-the-art sinogram-inpainting techniques. In particular, we demonstrate a 7 dB Peak Signal-to-Noise Ratio improvement compared to other methods.

**Keywords:** X-ray computed tomography; machine-learning; computer assisted design data; Generative Adversarial Network

## 1. Introduction

X-ray Computed Tomography (XCT) is a versatile 3D imaging technique that allows the estimation of volumetric X-ray attenuation profiles. The technique is used routinely in medical diagnosis but is also increasingly popular in scientific investigations and non-destructive testing [1,2]. However, in many applications, we can only collect limited X-ray measurements. For example, in industrial applications, object geometry might prevent full object rotation and thus prevent the use of standard XCT scanning strategies. The method might also be applicable to high-resolution medical scanning, where only a low-resolution full scan is possible to provide overall shape information, whilst a high-resolution scan can only be performed over a limited angular range.

Standard XCT reconstruction methods rely on measuring X-ray absorption along sufficiently many paths through an object. For example, for parallel beam tomography, measurements in a plane orthogonal to the rotation axis can be described as

$$R(\theta, r) = \int_x \int_y f(x, y) \delta(x \cos(\theta) + y \sin(\theta) - r) \, dx \, dy, \tag{1}$$

where $r$ is the distance of the measurement along the detector, $\theta$ the angle at which the measurement is performed, $(x, y)$ the coordinates of the object and $f$ the X-ray attenuation

of the object. $R(\theta, r)$ is known as the sinogram of the image $f(x, y)$ and the tomographic reconstruction problem can be posed as estimating $f(x, y)$ from a set of measurements $R(\theta, r)$. This 2D scanning setting can also be extended to 3D scanning. For example, in industrial settings, XCT systems often use a 3D cone beam scan trajectory; here, working directly on the full 3D scan data is crucial. Alternatively, where X-ray scattering is a problem, collimated fan-beam systems are also often used. Whilst these require longer scan times, they lead to decoupled inverse problems where each 2D image can be recovered independently.

Unfortunately, the tomographic inverse problem is severely ill-posed, even if enough angular measurements are available. Regularising the inverse problem is thus crucial, especially working in settings where only limited projections can be acquired. Standard image reconstruction in XCT relies on analytical methods based on the implicitly regularised Filtered Back-Projection (FBP) framework, initially developed for parallel beam tomography, but easily extendable to 2D fan-beam scan geometries by rearranging and interpolating the measurements. For 3D cone beam scans, the so-called FDK [3] algorithm provides a fast approximate analytical solution.

An alternative is to use iterative reconstruction methods. Whilst these offer more flexibility in terms of the constraints and regularisation approaches that can be used, given the size of the inverse problem, these methods remain computationally too costly for many practical applications. For example, many XCT systems routinely collect billions of measurements to reconstruct a 3D volumetric image with billions of unknown parameters. Fast and memory-efficient analytic approaches thus remain popular in practice.

When working with missing data, iterative regularised frameworks are tempting, as they often produce better images and allow the easy use of additional prior information. However, these are mainly applied to parallel or fan-beam scans, though advanced GPU accelerated methods are slowly allowing increased use of these approaches for 3D cone-beam data [4]. These methods solve the ill-posed or underdetermined linear inverse problem $Ax = y$ using additional regularisation terms such as the Total Variation constraint [5] or more XCT-specific regularisers [6–8]. Unfortunately, for real 3D cone beam CT applications, these methods remain too slow to be of practical significance. Thus, there remains significant interest in developing efficient approaches that can estimate the missing information required by analytical reconstruction methods [9,10].

In order to develop efficient methods for XCT image reconstruction, we here develop an approach that tries to estimate missing measurements. We infer the missing information in the sinogram and perform the inference using additional shape information. Then, standard analytical methods can be used to estimate the image. Whilst it would also be possible to incorporate shape priors into an iterative image recovery framework, for example, using masks that set attenuation values outside of objects to zero, for many realistic applications, this approach is still too slow for many problems of interest.

### 1.1. Related Work

To the best of our knowledge, there are currently no known published methods that use shape information to infer missing data for XCT reconstruction. There are, however, several techniques that are related to our sinogram estimation approach but that do not utilise shape information.

Earliest attempts on 2D sinogram data encode the sinogram as an image and missing measurements as zero-columns and apply various models, such as diffusion [9] or grid-interpolation [11], to fill out the missing values in the image. That is, given a sinogram displayed as an image, interpolating the missing pixel values in the image using a diffusion model or a grid-interpolation model. Tackling the problem of inferring measurements from that perspective is similar to the one of inpainting an image, that is, inferring arbitrarily large regions in images based on image semantics.

From a similar perspective, recent approaches [12,13] use data-driven techniques to improve sinograms. Their approach relies on initial deterministic interpolation to estimate the missing measurements. Then, they patch the sinogram into fixed blocks and

train a Convolutional Neural Network (CNN) to change all the true and interpolated values. Refs. [14,15] use a Unet [16] instead of a simple CNN. One must stress that these approaches aim at inferring measurements in scenarios where missing measurements are evenly spaced, which is different from the problem we chose to tackle. Other methods using Generative Adversarial Networks (GANs) [17] try to encode a latent representation of the few-view sinogram [18,19]. This method does not scale to reasonable image sizes (no more than $128 \times 128$ pixels). A GAN architecture has also previously been used for sinogram synthesis [20], with an optimisation procedure to perform the inpainting as in [21], though again without the use of shape priors.

Our proposed approach is based on semantic image inpainting, which is a constrained image generation problem [21]. Missing parts of an image are inpainted using a generative network and solving an optimisation problem. GAN and their fully-convolutional version Deep-Convolutional GAN (DCGAN) [22] are adapted to this task: they were used for image [23] and sinogram [20] inpainting. The optimisation relies on finding the "closest" encoding in the latent space of the GAN distribution by minimising a penalty function that encompasses contextual and conditional information. This method has two limitations. First, walking the latent space of the distribution can only yield specific improvements in the image generated by the GAN: ref. [24] shows that images can only be transformed to some degree (brightness, zoom, rotation). In this context, walking refers to applying small perturbations to the data's encoding in the GAN's latent space. Not only is the transformation corresponding to inpainting not defined, but it has no certainty to be achievable by applying some transformation to the latent representation, mainly when guided by a generic loss function rather than by a supervised walk. Second, the optimisation function of the inpainting procedure adds computational time and hyperparameters. In addition to training the generative model, the optimisation process is time-consuming and requires fine-tuning the learning rate and the number of iterations.

We propose an approach that uses shape information to side-step the complicated optimisation process of GAN-based inpainting methods.

### 1.2. Utilising Object Shape Information

Shape information is often available in medical and industrial imaging but rarely used in XCT reconstruction. For instance, in medical imaging, refs. [25,26] demonstrate the potential use of shape models of a generic human body to minimise X-ray exposure. Moreover, recent advances in anatomic modelling such as [27] demonstrate the growing availability of computational, yet accurate, priors. We want to stress that a generic shape prior would likely have limited benefits, so a patient-specific prior might be required. Such information could, for example, be obtained from faster, lower-resolution CT scans, such as in [28,29], or from different modalities, such as MRI.

In the field of non-destructive testing for high value manufacturing, Computer Assisted Design (CAD) drawings are often available, providing strong constraints on object shape. Frequently, when using CT scans to ensure the highest quality of manufacturing, the commercial entity that commissions the scan also possesses the CAD data related to the object that is scanned.

## 2. Proposed Approach

In an XCT system, measurements are taken at different rotation angles. We here address the problem of limited angle tomography, where X-ray measurements are not available from a (possibly extensive) range of consecutive angular directions(note that the limited angle setting here is different from a limited measurement setting where reduced numbers of measurements are collected at evenly or randomly distributed angles). The problem of inferring these missing measurements can thus be cast as an image inpainting problem [21]. If additional knowledge about the object's shape is available, the problem becomes an image inpainting with edge information task, which has already been studied in [30]. We thus here use the pix2pix Generative Adversarial Network (GAN) [31] archi-

tecture proposed in [30]. Whilst this generic inpainting approach has been proposed and demonstrated for image inpainting with additional shape information, it has never been used for XCT sinogram estimation. The generic idea is shown in Figure 1.

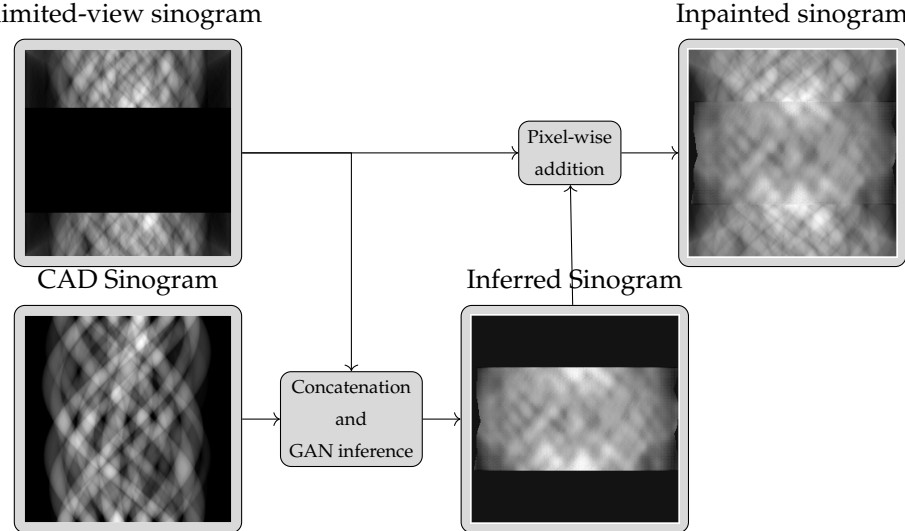

**Figure 1.** The pix2pix architecture using a generator and a discriminator. The generator is trained to generate missing sinogram information from a multi-channel input. One channel is the sinogram with the missing information, one is a sinogram encoding the object shape and one is a mask indicating where the CAD expects missing information to be located. The discriminator (or detector) is a classification network that tries to detect whether the multi-channel input is an actual full sinogram or is a sinogram where the generator estimates missing information. GAN training is done iteratively, with the generator trying to fool the discriminator better and the discriminator trying to improve the detection of estimated data.

We propose using the pix2pix network architecture to incorporate CAD information into the XCT inpainting processes. We develop the method using a 2D/3D parallel beam setting, which can easily be extended to 2D fan-beam scans. For 3D cone-beam data, the same approach can be used in theory on the entire set of projections, though here a block-based approach might be required to reduce computational requirements. We show how measurement information from limited angles can improve the initial shape prior.

### 2.1. Generative Adversarial Networks for Sinogram Inpainting with Edge Information

The pix2pix architecture uses a GAN approach. GANs are generative models that try to learn complex probability distributions by providing an explicit method to generate samples from that distribution. GANs comprise two networks that compete against each other. A generator $G$ generates new data that follow the same distribution $p_{data}$ as the training samples by transforming a noise vector drawn from a distribution $p_z$. The discriminator $D$ tries to discriminate between real training samples and those generated by $G$. They are a powerful alternative to other deep generative models and density estimators.

GAN optimisation is done by alternatively solving the minimax problem:

$$\min_G \max_D \mathcal{L}(G, D) = \mathbb{E}_{x \sim p_{data}} \log(D(x)) + \tag{2}$$

$$\mathbb{E}_{z \sim p_z} \log(1 - D(G(z))), \tag{3}$$

where $\mathbb{E}$ is the expectation over the training data.

#### 2.1.1. Deep Convolutional GAN and U-Net—The pix2pix Architecture

A problem commonly found in GANs is instability during training. To address this problem and use convolutions to scale to larger data sizes, the Deep Convolutional GAN

(DCGAN) architecture [22] was proposed. Other changes proposed include the replacement of the maxout [32] activation by ReLu and Tanh in the generator and LeakyReLu in the discriminator, the inclusion of batch-normalisation [33] and the replacement of pooling units with learnt sampling units. The pix2pix architecture uses a U-net with several input channels as the generator to include prior information. In our setting, one channel is the measured sinogram, where missing information is set to zero, a second channel encodes the shape information and a third channel provides a mask to tell the network which information is to be estimated and which information is given. An example of a target sinogram and the three different input sinograms to our generator model are shown in Figure 2 for one of the datasets introduced below. U-nets were initially designed for image segmentation and are fully convolutional auto-encoders which use the feature channels inferred from a down-sampling step to a latent representation to up-sample this latent representation back to the original space. They are convenient architectures designed to map a multi-channel input image to a one-channel output image or vice versa. U-nets have been shown repeatedly to outperform other architecture on many tasks and have thus developed as a de facto standard in many image processing tasks. The training procedure is detailed in Figure 3.

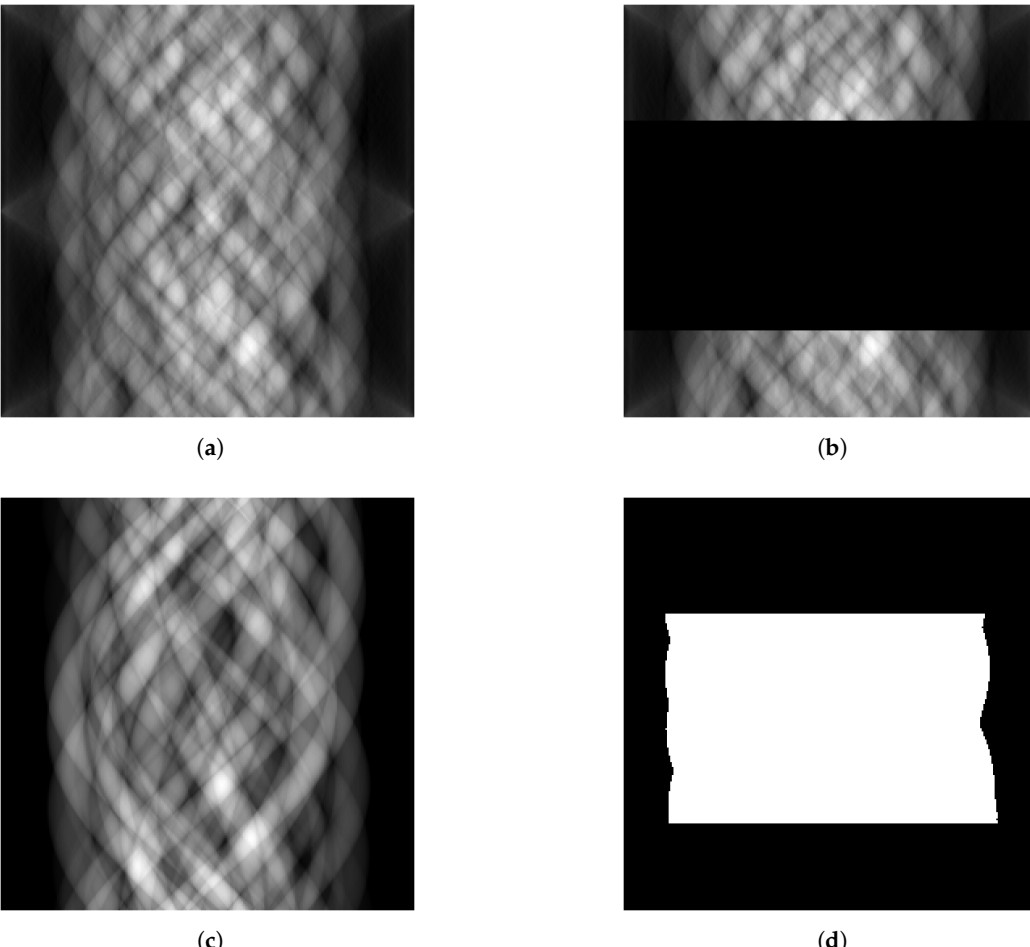

(a)

(b)

(c)

(d)

**Figure 2.** Sample of the input data given to the generator. We estimate the full sinogram (**a**) from the sinogram with missing data (**b**), the sinogram of the shape prior (**c**) and the sinogram's mask indicating the location of missing projections (**d**).

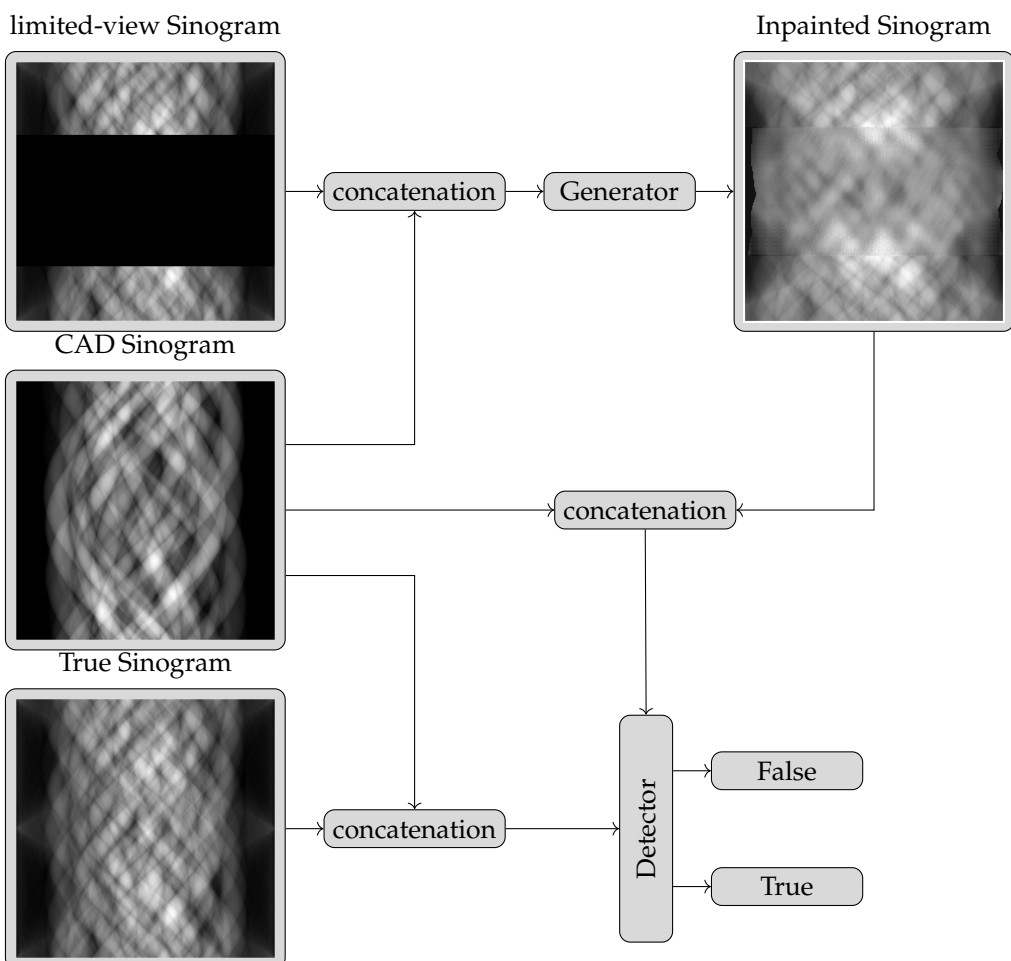

limited-view Sinogram

Inpainted Sinogram

CAD Sinogram

True Sinogram

concatenation → Generator

concatenation

concatenation

Detector → False

Detector → True

**Figure 3.** Explanation of the training procedure. The CAD sinogram is added to both target and limited-view sinograms.

### 2.1.2. Adapted Loss Function

The standard GAN loss function in Equation (2) does not account for conditional modalities. Furthermore, in order to encourage the generator to produce images close to their target value, an L1-loss is often used [21] and is also a good choice for image quality when Poisson noise is present, as is the case in XCT images [34]. The pix2pix architecture is thus trained using the loss function:

$$\min_{G} \max_{D} \mathcal{L}(G, D) = \mathbb{E}_{x \sim p_{true}} \log(D(x)) + \tag{4}$$

$$\mathbb{E}_{\tilde{x} \sim p_{impaired}} \log(1 - D(G(\tilde{x}, y))) + \tag{5}$$

$$\lambda L_1(G(\tilde{x}, y), x), \tag{6}$$

In our case, $x$ and $\tilde{x}$ are the target sinograms with all data and the input sinogram without the missing data, respectively. $y$ is an image encoding the prior shape and $\lambda$ is a weighting parameter.

The problem that the discriminator has to solve is as follows: given a full sinogram and its associated shape prior, classify the sinogram as either generated by the GAN or drawn from the authentic database. Given a sinogram with missing data, the generator has to generate missing acquisitions that are close to the ground-truth acquisitions in terms of L1-loss [35].

## 2.2. Encoding the Shape Prior

There are several ways to encode the object's shape information for use in the GAN. We work in the sinogram domain, encoding shape priors in the same domain. This will be done by projecting images containing shape information into the sinogram domain. One way to generate the shape image would be to generate images where the boundary pixels of an object are set to a non-zero value. Alternatively, we can set the object boundary and interior pixels to a single value before projection, using the same value for all objects or choosing a random value for each object. We here chose the latter approach as it allows a greater range of different densities to be encoded and was found to work better in initial experiments. An example shape prior encoding is shown in Figure 4, where we show the original image and the shape prior.

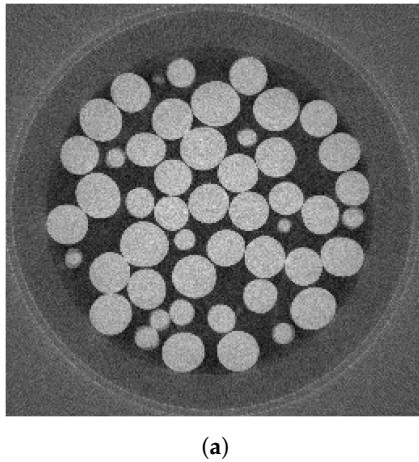
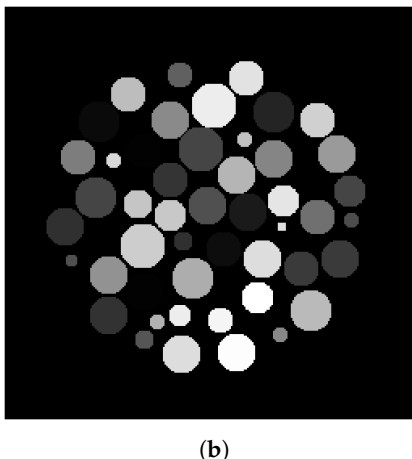

(**a**)                    (**b**)

**Figure 4.** Real training samples. A 2D slice from the SophiaBeads dataset, showing the full slice we want to estimate (**a**) and the image that encodes the shape prior (**b**). Here the shape prior indicates each of the locations for each sphere in the image. However, as the attenuation value of each sphere is unknown, random attenuation values are assigned.

To further enhance the encoding of the prior, we add a further pre-processing step that uses a property of sinograms. For XCT scans for which the entire object remains in the field of view, at any acquisition angle, when summing all attenuation values, the total attenuation should be constant, independent of projection angle:

$$\int_r R(\theta, r) = \int_y \int_x f(x, y) \, dx \, dy = \text{const.} \tag{7}$$

We use this property to scale the shape encoding values for each sinogram angle. For region-of-interest scans where objects can move outside the field of view, interpolation could be used to estimate this value for each missing angle, though we do not explore this here.

### Architectural Details

The architecture of our generator follows the guidelines of the pix2pix architecture, and our description follows the same notation. Let $C_k$ be a Convolution-BatchNorm-ReLU layer with $k$ output channels. Let $CD_k$ be a Convolution-BatchNorm-Dropout-ReLU layer with a dropout rate of 50% [36]. All convolutions are four-by-four spatial filters applied with a stride of 2. Convolutions in the encoder down-sample by a factor of 2, whereas in the decoder, they up-sample by a factor of 2. After the last layer in the decoder, a convolution is applied to map to the number of output channels (which in turn depends on how we chose to encode data), followed by a Tanh function and an element-wise multiplication with a mask, as explained in Section 2.2. As an exception to the above notation, BatchNorm is not applied to the first C64 layer in the encoder. All ReLUs [37] in the encoder are

leaky [38], with slope 0.2, while ReLUs in the decoder are not leaky. This results in the following generator:

Encoder:

$C_{64}$-$C_{128}$-$C_{256}$-$C_{512}$-$C_{512}$-$C_{512}$-$C_{512}$-$C_{512}$.

Decoder:

$CD_{512}$-$CD_{1024}$-$CD_{1024}$-$C_{1024}$-$C_{1024}$-$C_{512}$-$C_{256}$-$C_{128}$.

The discriminator works on image patches of size $16 \times 16$. For the discriminator, after its last layer and after extracting the patch, a convolution is applied to map to a one-dimensional output, followed by a Sigmoid function. This results in the following discriminator:

Discriminator:

$C_{64}$-$C_{128}$-$C_{256}$-$C_{512}$.

## 3. Datasets

Deep-Learning is a data-driven paradigm that requires large datasets for training. However, databases of XCT acquisitions with associated shape priors do not exist. We, therefore, adapted the SophiaBeads [39] dataset to report the performance of our process on actual data.

### 3.1. Preparing the SophiaBeads Training Data

Initially designed to provide data for researchers into XCT reconstruction from limited measurements, this dataset is a collection of six cone beam XCT acquisitions of a plastic tube filled with soda-lime glass beads. As the beads are approximately cylindrical, each slice through the volume contains only circular objects. The difference between each dataset is the number of projections used per volume.

To generate our training data with both complete and missing projections as well as shape information, we reconstructed the two datasets that had 512 and 1024 projections using the approach in [4,40], generating $256 \times 256 \times 200$ volumes, using the FDK algorithm for cone-beam tomography. The other four datasets needed more projections to provide suitable ground-truth training data and were thus discarded. We used circle-finding methods to identify object boundaries in the full reconstructions, which were used as shape priors. Training data were then generated from individual 2D slices using a parallel beam geometry for experimental convenience and speed.

Thus, our datasets consisted of 400 sinograms (200 generated from each of the two volumes) and the associated CAD data. Each volume is split into 180 training slices and 20 test slices. A slice of the volume reconstructed with 512 projections and the identified beads, from which the object boundaries can be inferred, is shown in Figure 4a and Figure 4b, respectively, whilst the associated sinograms were those previously shown in Figure 2. To test our method's robustness, we chose not to capture the plastic container's outline in the CAD data, and several small circles were also not included. Doing so ensured the images had features not represented in the CAD data.

We generate full sinograms from each slice, simulating a detector with 256 pixels and measuring from 256 equally spaced angles in a 180-degree arc. We use the implementation of the radon transform proposed in [4]. The sinograms corresponding to Figure 4a and Figure 4b are shown in Figure 2a and Figure 2c, respectively. Sinograms with missing projections were generated by setting between 5 and 95% of consecutive angular measurements to 0. Figure 2b shows an example of a sinogram with 50 per cent missing data.

### 3.2. Synthetic Training Data

Having a synthetic dataset allows experimentation with the data model. In real applications, the shape prior will only sometimes precisely match the object shape. There could be boundary deviations, or objects could have unknown inclusions and internal

defects. Therefore, we also generate a separate dataset with additional internal defects in the objects not captured by the shape prior.

In order to simulate such defects, we create a synthetic dataset to mirror the SophiaBeads data, as shown in Figure 5a,b. Each slice is a 256 by 256 image of several circles generated with the circle function from the scikit-image.draw Python module with different radii and positions and uniform densities with added Gaussian noise. Object attenuation values are set to 25, while the image background is assumed to have an X-ray attenuation much lower than the object and is set to 1. We generate the shape prior associated with a slice using the same function with the same radii and position for each circle but with different densities and no Gaussian noise. Whilst we chose this model to replicate the SophiaBeads dataset, similar data have previously been used in [41–43]. Figure 6 displays an example image slice.

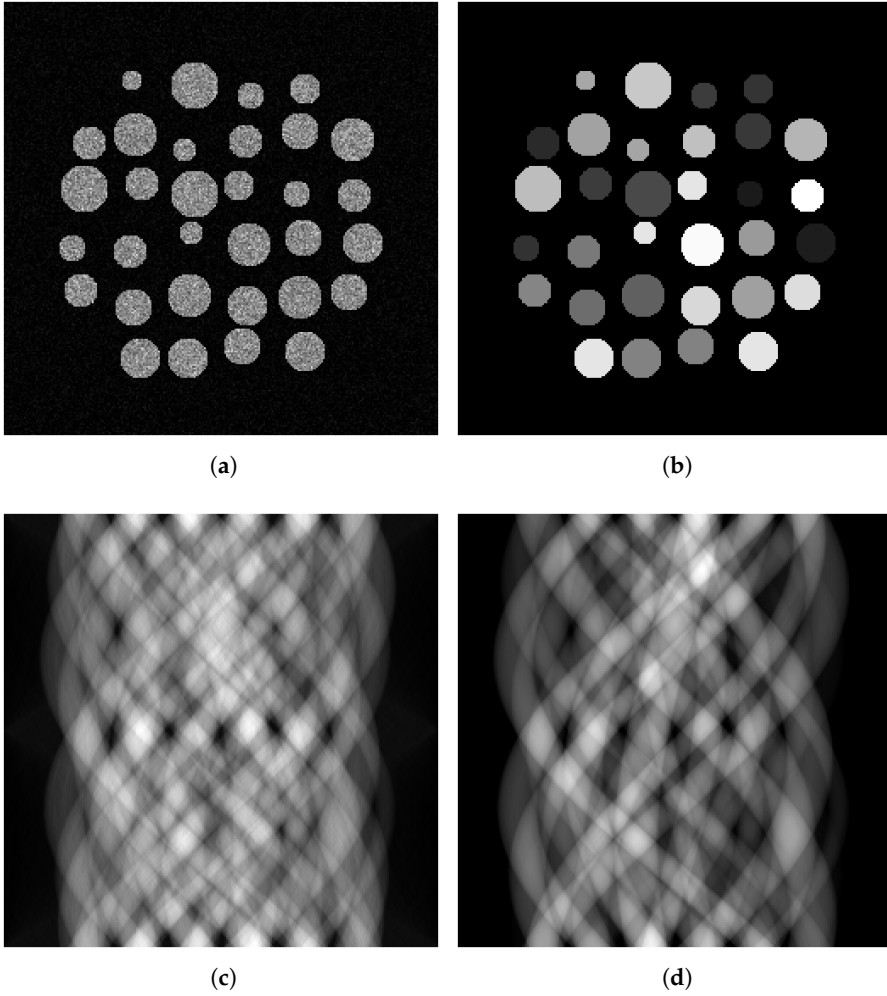

(a)                                                                                                    (b)

(c)                                                                                                    (d)

**Figure 5.** Slice samples from the synthetic dataset. The ground-truth reconstruction (**a**), and the shape prior (**b**) showing each object with a different randomly assigned grey value, in which no plastic container is visible. In an attempt to reproduce noise in the image, Gaussian noise is added to the material of uniform densities. (**b**) Sinogram samples from the synthetic dataset, from the ground-truth reconstruction (**c**) and from the shape prior (**d**). The observed sinograms are obtained using the forward radon transform with a parallel beam geometry.

### 3.3. GAN Training

We train the pix2pix networks on the SophiaBeads training dataset reconstructed with 1024 projections using a hundred epochs. We then evaluate them on the dataset reconstructed from 512 projections. The three-channel input data are re-scaled to have values between 0 and 1. We used a batch size of 8 and the Adam optimiser [44]. We set

the learning rate at 0.0002 and $\lambda$ to 100. We used label-smoothing and noise addition in the discriminator, as suggested in [31]. We implemented the top-k [45] training procedure which discards 6/8 of the generated samples when updating the *G* loss.

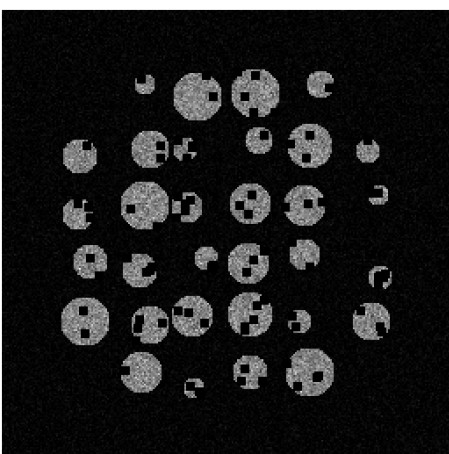

**Figure 6.** Simulated XCT images with objects with internal defects that are not encoded in shape prior.

We perform training on the SophiaBeads dataset with the priors and on a synthetic dataset that models internal defects separately. This means two copies of the same architecture were trained separately on each dataset but with the same hyperparameters.

## 4. Results

Whilst no known methods incorporate shape information into XCT reconstruction, we propose to compare our approach to a set of techniques that address the problem of inferring missing measurements in a sinogram.

First, we propose one straightforward method given CAD data: replacing the missing projections with the ones expected by the CAD. We also implement scaling the CAD data values developed in Section 2.2. These two CAD-only approaches are reported in rows one and two of Table 1. Then, we also implement a linear interpolation method. This straightforward technique is used as a pre-processing step in many interpolation procedures, such as in [13]. In row three of Table 1, we refer to this method as "Linear Interpolation".

**Table 1.** Performance comparison between the different methods regarding the difference between the estimated and true sinogram.

| Missing Angles | 30° | 60° | 90° |
|---|---|---|---|
| (1) Replacing missing projections with CAD data | 22.5 | 19.2 | 17.1 |
| (2) Replacing missing projections with CAD data (and with attenuation values scaling) | 23.1 | 21.3 | 19.7 |
| (3) Linear Interpolation | 23.6 | 22.6 | 20.0 |
| (4) Interpolation using a Unet | 9.9 | 8.4 | 7.7 |
| (5) Interpolation using a Unet and CAD data | 16.8 | 12.9 | 11.0 |
| (6) Interpolation using a GAN | 7.0 | 3.82 | 2.0 |
| (7) Interpolation using pix2pix and CAD data | **29.1** | **27.8** | **27.0** |

In order to compare the difference between generative models and their non-generative counterparts, we implement two methods based on the Unet architecture. To begin with, we train a Unet to infer missing projections from the impaired sinogram only. Then we train a Unet with the same architecture to tackle the same goal, but we provide it with shape information. This comparison will also hint at the importance of shape priors for tackling the problem of inferring missing projections in a sinogram. These two methods are referred to as "Interpolation using a Unet" and "Interpolation using a Unet and CAD data" in Table 1. They are given in rows four and five, respectively.

We also compare our method to the GAN-only approach developed in [20]. We refer to this method as "Interpolation using a GAN" in Table 1, in row six. Unfortunately, we did not manage to replicate their results on our dataset. We suspect this might be due to an increased sinogram size. Finally, we report the results of our method based on the pix2pix network and refer to this method as "Interpolation using pix2pix and CAD data" in row seven of Table 1.

We report the results using the Peak Signal-to-Noise Ratio (PSNR).

### 4.1. Estimation Performance on the SophiaBeads Dataset

We have two volumes of 200 slices each. We split each volume into 90 per cent for training and 10 per cent for testing. We train using only the data from one volume, with convergence monitored on the same volume's test data. We then compute the performance results presented below by using the test split from the other volume.

We start by comparing the approaches over a range of misusing angles, from 30 degrees (17% missing data) to 90 degrees (50% missing data). In particular, we compare the method that estimates missing information from the CAD data with and without scaling, simple linear interpolation of the sinogram, interpolation followed by a U-net, interpolation followed by a U-net that also sees the CAD data, GAN-based inpainting without CAD data and finally the pix2pix U-net-based GAN that also uses CAD data. We evaluate the performance by comparing the estimated sinogram to the target sinograms.

Table 1 shows the results, where we can see that the pix2pix architecture combined with our CAD prior outperforms all other approaches in terms of PSNR. It is interesting to note that using the CAD prior significantly enhances the performance of the U-net. It is also worth mentioning that the scaling operation slightly improves the quality of the CAD replacement. The GAN method was hard to train. Hence, the inpainting process has limitations that we will detail in Section 5.

Sinograms contain more low-frequency information than images, so the error measure in the sinogram domain differs from the error in the image domain. To evaluate the performance of the different methods once we have recovered the image, we compute a reconstruction using the SIRT algorithm as implemented in the TIGRE Python toolbox [4].

Figure 7 shows the reconstructions associated with each method when 50% of the data is missing. As can be seen in Table 2, not only does the pix2pix architecture combined with the CAD prior outperform all other approaches in the image space, but it is the only one that yields an improvement compared to the reconstruction made from the original sinogram as shown here in the first row of the table. It is also important to underline that methods performing well in the sinogram space do not necessarily improve the image space.

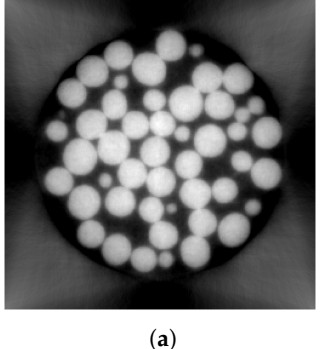 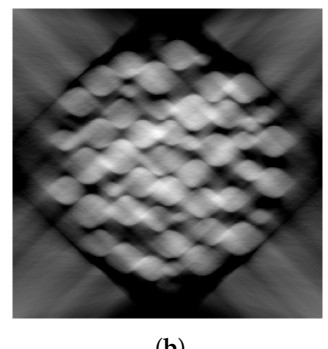 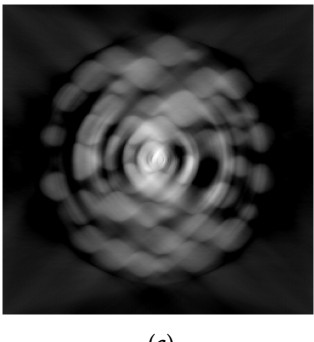

(a) (b) (c)

**Figure 7.** *Cont.*

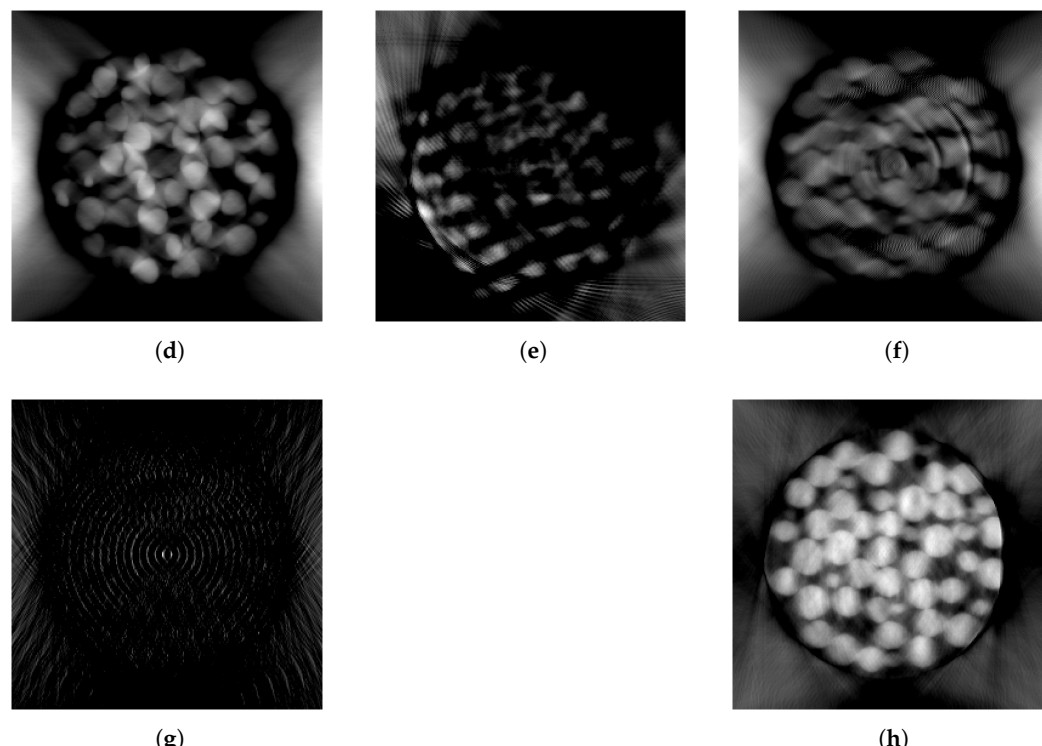

**Figure 7.** Reconstructions of a slice from one of the SophiaBeads test datasets. First, the target image (**a**), reconstructed from 256 measurements. Then, the image reconstructed from half of the measurements (**b**), the linear interpolation (**c**), the replacement of the missing measurements by the ones expected by the CAD (**d**), the measurements inferred by the Unet method without CAD data (**e**), the measurements inferred by the Unet method with CAD data (**f**), the measurements inferred by the GAN without CAD data (**g**) and the measurements inferred by our method (**h**).

**Table 2.** Performance of the different methods in terms of the quality of the reconstructed image.

| Missing Angles | 30 ° | 60° | 90° |
|---|---|---|---|
| Reconstruction from sinograms without treatment | 69.1 | 66.4 | 63.8 |
| Reconstruction corresponding to Table 1 (2) | 61.3 | 58.2 | 56.3 |
| Reconstruction corresponding to Table 1 (3) | 61.2 | 61.8 | 59.4 |
| Reconstruction corresponding to Table 1 (4) | 52.7 | 51.4 | 51.4 |
| Reconstruction corresponding to Table 1 (5) | 60.4 | 56.5 | 55.1 |
| Reconstruction corresponding to Table 1 (6) | 53.4 | 51.4 | 50.9 |
| Reconstruction corresponding to Table 1 (7) | **69.6** | **67.9** | **66.5** |

### 4.2. Effect of Inconsistencies between the Two Modalities on the Synthetic Dataset

We finally also use the data with simulated defects. The advantage of our method here is less pronounced, and a qualitative inspection shows that the network fails at inferring the hole pattern, thus showing a limited ability to recover defects. This can be seen in Figure 8, where we show four examples of the difference between the image reconstructed with our method and the target image for data with 33% per cent missing projections. This suggests that the GAN might fail to learn the fine structure of the object attenuation.

One reason for this might be that we are estimating a considerable number of missing projections, which is highly challenging. Moreover, the nature of holes makes the choice of loss function delicate, as the defects are high-frequency image components whilst low-frequency information dominates. In a future study, we plan to modify the loss function to provide more appropriate weighting to the cost function. The image reconstruction process

treats measured and inferred data similarly, which might not be optimal if one or the other is known to have a more significant error.

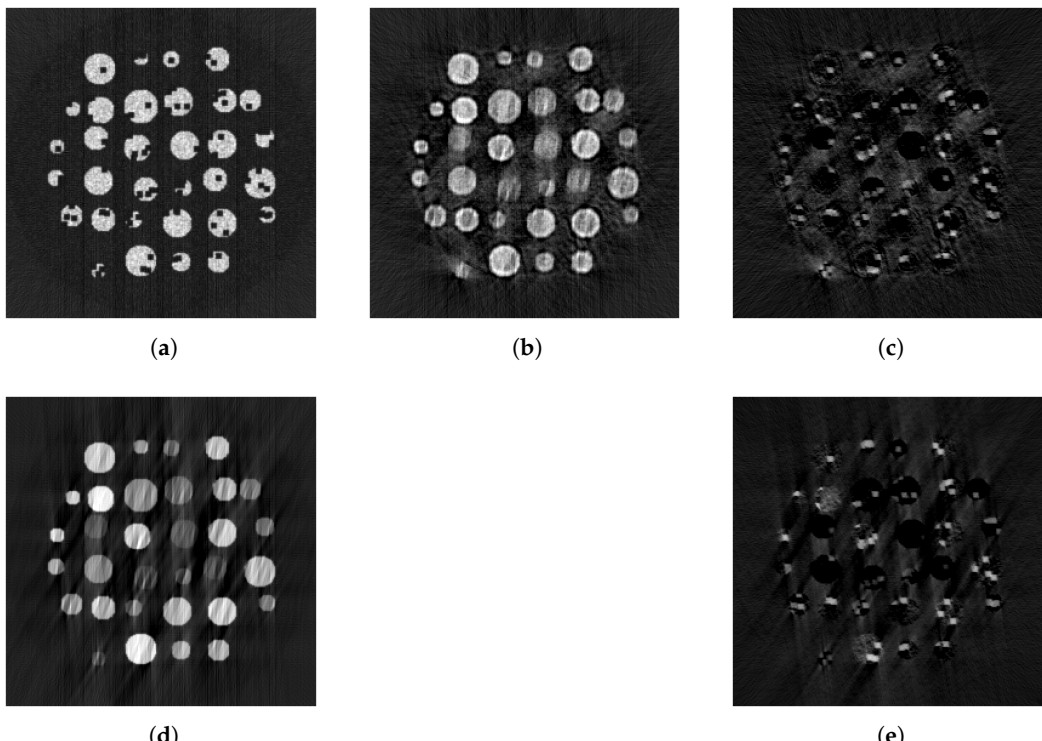

**Figure 8.** Reconstructions of the synthetic samples using various methods. First, the one from the sinogram 256 measurements (**a**), then from the sinogram interpolated with our method (**b**), the difference between the latter and the target (**c**), the one interpolated with the shape prior (**d**) and the difference between the latter and the target (**e**). When reconstructing data with defects using shape priors, the method can identify the objects' location, but the defect's fine details are more difficult to identify.

## 5. Discussion and Conclusions

In this paper, we have shown how to use a GAN to exploit shape prior information to address the image inpainting with edge information problem in XCT. Our experiments show the significant advantage that our method offers over state-of-the-art methods that do not include shape prior information for inferring a large number of consecutive missing acquisitions. We also demonstrated that the CAD prior facilitates GAN training.

The main limitation of our approach is the failure to extrapolate faults that are not encoded in the shape priors. One reason might be that our cost function does not sufficiently account for high-frequency image information. Moreover, whilst the image reconstruction process is not studied here, weighting the ground truth and the generated acquisitions could improve the final image quality.

Comparing our method to the GAN-only approach of [20], it is worth stressing that the sinograms that are measured in industrial applications are much larger than those used in their report and so scaling of the approach has to be considered. The method in [20] was directly inspired by [23], where results are achieved on 64 × 64 images, and learning is based on contextual as well as discriminator losses. This approach failed because it did not directly scale to the increased data sizes used here.

In a future study, we will investigate the mitigation of the impact of the discrepancies between the CAD and the actual scan. To do so, we plan to add random image perturbations to the CAD before sampling. We hope that this approach will help the model learn more robust image features.

**Author Contributions:** Conceptualization, E.V. and T.B.; methodology, E.V. and T.B.; software, E.V. and T.B.; validation, E.V.; formal analysis, E.V. and T.B.; investigation, E.V. and T.B.; resources, T.B.; data curation, E.V. and T.B.; writing—original draft preparation, E.V.; writing—review and editing, E.V., K.F. and T.B.; visualization, E.V.; supervision, K.F. and T.B.; project administration, T.B.; funding acquisition, T.B. All authors have read and agreed to the published version of the manuscript.

**Funding:** This research was funded by Anglo-French DSTL-AID Joint-PhD program.

**Institutional Review Board Statement:** Not applicable.

**Informed Consent Statement:** Not applicable.

**Data Availability Statement:** The data used in this study is the SophiaBeads dataset, that can be downloaded from Zenodo. We refer the reader to [39] for the dataset and to [40] for the code to reconstruct the tomographic images.

**Conflicts of Interest:** The authors declare no conflict of interest.

## Abbreviations

The following abbreviations are used in this manuscript:

| | |
|---|---|
| XCT | X-ray Computed Tomography |
| CAD | Computer Assisted Design |
| (DC)GAN | (Deep-Convolutionnal) Generative Adversarial Network |
| CNN | Convolutionnal Neural Network |

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
