# Peer review of "Sinogram Inpainting with Generative Adversarial Networks and Shape Priors"

_tomography, doi:10.3390/tomography9030094_

Round 1

Reviewer 1 Report

This work presented a method with GAN and shape prior information to generate missing part of Xray-CT sinogram. Specifically, it targeted consecutively missing data. The introduction is thorough, and the method is clearly stated with all hyperparameters given for easy replication. The advantages and limitations are discussed. The paper is well written. The topic is of interest and the results are worthwhile for publication and further exploration. I thus recommend its publication in this journal yet with some revision suggestions for possible improvement of the paper quality and clarification of some doubts.

1, page 6, line 171. Should it be “input sinogram with missing data”?

2, Comparison of different methods should be clearer. I knew line 305 to line 309 should be matched to the first paragraph of Results. However, they are quite confusing given their names are not consistent. For instance, “the method that estimates missing information from the CAD data with and without scaling”. I did not find any place, stating how to estimate missing information. Besides, there are some inconsistences between table 1 and table 2. Is “non-scaled CAD” in table 1 the same as
direct recons” in table 2? Perhaps the authors could consider use one paragraph to name or label these methods as well as illustrating what the authors did.

3, Evaluation metric should be used more carefully. First, PSNR and SSIM should be described clearly. Are they evaluated on the whole image or block by block followed with an average process? The authors may need to provide PSNR and SSIM only in the center part as well. It is ridiculous that Figure 8g and 8a have SSIM of 0.99. This either proved the evaluation by SSIM is meaningless or the way to use it is doubtful. Second, the absolute value (HU) of a CT reconstructed image has physical meaning. I think it is better to show comparison between profile values along the center line.

4, typo. Page 11, line 297. Peak Signal to Noise Ratio.

5, what if the boundary is not so obvious as shown in figure 4a? For example, you can only get shape information from low resolution figures, for example Gaussian convoluted figure 4a, as stated in the first paragraph of Introduction. Could the authors discuss it or show some results? This is of more interest in medical applications.

Author Response

  • 1, page 6, line 171. Should it be “input sinogram with missing data”? 

Response: Corrected

  • 2, Comparison of different methods should be clearer. 

Response: We agree that our exposition could have been more explicit, and we have now reworked the relevant sections and rewritten the specific paragraph highlighted in the review.

  • 3, Evaluation metric should be used more carefully. 

Response: We agree with both reviewers who commented on this. As the SSIM did not provide meaningful comparisons that relate to visual differences, we now concentrate on the PSNR measure in the revised manuscript. 

  • 4, typo. Page 11, line 297. Peak Signal to Noise Ratio.

Response: Corrected

  • 5, what if the boundary is not so obvious as shown in figure 4a? For example, you can only get shape information from low resolution figures, for example Gaussian convoluted figure 4a, as stated in the first paragraph of Introduction. Could the authors discuss it or show some results? This is of more interest in medical applications.

Response: This is indeed an interesting point that will warrants further investigation. We have now added a discussion about this to the paper but have to postpone a detailed experimental investigation to future work.

Reviewer 2 Report

The proposed approach is a good application of the GAN for the important problem in tomographic image reconstruction. The results appear to demonstrate that the method is on the right track, although much research is needed to make it practical.

1.    The introduction of “shape prior” is the key contribution of the proposed method. However, I had a hard time how such image data are obtained using CAD. If the imaging subject is a standard phantom, certainly we can generate the image with CAD, and can be used with GAN. However, if a subject with a wide variety of shapes and materials is being imaged, how can one generate the CAD data?

2.    Figure 2 can be removed without losing the clarity of the authors’ message. After all, it is not referred to in the main text.

3.    The equation in line 149 needs the equation number “(1)”.

4.    What are “two volumes” in line 238? Those may be “complete” and “missing” projections in line 230. However, for the missing projections, there are at least three sets 30, 60, and 90 degrees.

5.    The paragraph of lines 294-298 belongs to the method section.

6.    Table 2 was not referred to in the main text. Table 1 in line 322 must be Table 2.

Author Response

  • 1. However, if a subject with a wide variety of shapes and materials is being imaged, how can one generate the CAD data?

Response: The use of shape priors is central in our paper, and our use of the term CAD data was inspired by the potential use of the method in industrial applications. In medical applications, a generic shape prior would likely have limited benefits, so a patient-specific prior might be required. Such information could, for example, be obtained from faster, lower-resolution CT scans or from different modalities, such as MRI. We have now re-worded the paper to make this clearer. 

  • 2. Figure 2 can be removed without losing the clarity of the authors’ message. After all, it is not referred to in the main text.

Response: corrected.

  • 3. The equation in line 149 needs the equation number “(1)”.

Response: corrected

  • 4. What are “two volumes” in line 238? Those may be “complete” and “missing” projections in line 230. However, for the missing projections, there are at least three sets 30, 60, and 90 degrees.

Response: We agree that the term “two volumes” was misleading. We have re-worded this section to make our meaning more straightforward.

  • 5. The paragraph of lines 294-298 belongs to the method section.

Response: corrected

  • 6. Table 2 was not referred to in the main text. Table 1 in line 322 must be Table 2.

Response: corrected